# Dietary Evodiamine Inhibits Atherosclerosis-Associated Changes in Vascular Smooth Muscle Cells

**DOI:** 10.3390/ijms24076653

**Published:** 2023-04-03

**Authors:** Yiwen Zha, Yongqi Yang, Yue Zhou, Bingqian Ye, Hongliang Li, Jingyan Liang

**Affiliations:** 1Department of Human Anatomy, Histology and Embryology, Medical College, Yangzhou University, Yangzhou 225001, China; 2Institute of Translational Medicine, Medical College, Yangzhou University, Yangzhou 225001, China; 3Jiangsu Key Laboratory of Integrated Traditional Chinese and Western Medicine for Prevention and Treatment of Senile Diseases, Yangzhou University, Yangzhou 225001, China

**Keywords:** evodiamine, VSMCs, atherosclerosis, proliferation, inflammation

## Abstract

*Evodia rutaecarpa (Juss.) Benth* is a traditional Chinese medicine. The active ingredient, evodiamine, is a quinolone alkaloid and is found in Evodiae fructus. We investigated the effect of evodiamine on atherosclerosis using *LDLR−/−* mice fed on a high-fat diet and ox-LDL-induced MOVAS cell lines to construct mouse models and cell-line models. We report a significant reduction in atherosclerotic plaque formation in mice exposed to evodiamine. Our mechanistic studies have revealled that evodiamine can regulate the proliferation, migration, and inflammatory response of and oxidative stress in vascular smooth muscle cells by inhibiting the activation of the PI3K/Akt axis, thus inhibiting the occurrence and development of atherosclerosis. In conclusion, our findings reveal a role for evodiamine in the regulation of vascular smooth muscle cells in atherosclerosis, highlighting a potential future role for the compound as an anti-atherosclerotic agent.

## 1. Introduction

Atherosclerotic cardiovascular disease (CVD) remains a leading cause of vascular disease worldwide [1,2]. Under physiological conditions, vascular smooth muscle cells (VSMCs) mainly exist in the media layer of blood vessels, and their proliferation and migration can repair vascular damage [3]. However, during disease development, such as in atherosclerosis, migration of VSMCs from the media into the intima leads to an accumulation of smooth muscle cells in the growing atherosclerotic plaque. Over time, these cells proliferate and secrete extracellular matrix macromolecules which constitute the mass of an established atherosclerotic plaque [4,5,6].

Proliferation and migration of VSMCs is a highly dynamic and complicated process and is intricately regulated by multiple factors [7,8,9]. One key factor is the interaction of inflammatory and oxidative stress responses in VSMCs. Inflammatory cytokines promote the development and progression of inflammation and the proliferation and migration of VSMCs [10,11]. In addition, reactive oxygen species (ROS) are produced in response to chronic and mild inflammation, which further stimulates proliferation and migration of VSMCs through the activation of signaling pathways [12,13]. Therefore, inhibition of the proliferation and migration of VSMCs through reduced inflammation and oxidative stress represents an attractive therapeutic strategy to delay the progression of atherosclerosis.

In traditional Chinese medicine, Evodia rutaecarpa, a plant rich in evodiamine and rutaecarpine, has been extensively applied in Wu-zhu-yu decoction and Zuo-jin-wan in the treatment of migraine. Evodiamine is a quinolone alkaloid containing numerous biological effects [14], including anti-inflammatory [15], anti-tumor [16,17], and anti-obesity [18] properties. Evodiamine exhibits anti-tumor properties by primarily inhibiting the movement of various cancer cells. For example, evodiamine has been reported to inhibit the proliferation and invasion of bile duct cancer cells by upregulating *SHP-2* expression to inhibit the IL-6/STAT3 signaling pathway [19]. In addition, in non-small cell lung cancer, evodiamine increased CD8 T cells and downregulated the MUC1-C/PD-L1 axis [20]. In addition to its anti-tumor properties, evodiamine has a protective effect on the cardiovascular system. For example, treatment with berberine combined with evodiamine can prevent hyperlipidemia by inhibiting the intestinal absorption of cholesterol [21]. In addition, evodiamine has a vasodilatory effect in the isolated mesenteric artery of rats, which is endothelium dependent [22].

Although these findings suggest that evodiamine may have beneficial anti-tumor and cardio-protective properties, it remains unclear whether evodiamine can influence VSMC proliferation and migration by regulating inflammation and the oxidative stress response in VSMCs, thereby inhibiting the occurrence and development of atherosclerosis. This study attempted to demonstrate any anti-atherosclerotic effect of evodiamine and explore underlying molecular mechanisms in appropriate mouse and cell line models.

## 2. Results

### 2.1. Evodiamine Reduces Atherosclerotic Plaque in Mice

In order to determine the effect of evodiamine on the formation of atherosclerotic plaque lesions, 8-week-old male *LDLR−/−* mice were randomly divided into two groups and fed a high-fat diet for 8 weeks. In addition, one group was given evodiamine treatment at a dose of 10 mg/kg/d by daily gavage for 8 weeks. The other group was treated with saline as the control group. The aorta was studied to determine atherosclerotic lesion sizes and properties and compare between the evodiamine- and control-treated mice.

Quantitative analysis revealed that the lesion area on the aorta surface was significantly lower in the evodiamine-treated mice (control group 15.3%; evodiamine group 6.2%), compared with the control group (Figure 1A,B). H&E staining and Masson staining of the transverse section of the aortic root revealed that the thickness of atherosclerotic plaque in the aortic root was significantly reduced in the evodiamine group, and the proportion of the necrotic core was also greatly reduced. We found no significant difference in the proportion of collagen content between the two groups of mice (Figure 1C,E–G).

In addition to the aorta surface staining analysis, immunofluorescence was used to detect the expression of Myh11, a contractile marker, in the smooth muscle cells of the diseased aorta root in mice. As shown in Figure 1D, the number of VSMCs in aortic root plaques decreased significantly after evodiamine treatment. Therefore, it can be speculated that evodiamine may delay the atherosclerosis process by regulating VSMCs.

After confirming the efficacy of evodiamine at 10 mg/kg/d in impairing atherosclerotic plaque formation, we further investigated the toxicity of long-term use of evodiamine at this concentration in mice. To assess this hypothesis, 8-week-old male *LDLR−/−* mice were treated daily for 8 weeks, before liver and kidney function measurements at week 8. Mouse weight was recorded at weeks 2, 4, 6, and 8. The results of this analysis revealed no evidence of toxicity (Figure 1H–L). In addition, there were no significant differences in the appearance or H&E staining of the heart, liver, spleen, lungs, and kidneys between the evodiamine and control groups (Figure 1M,N).

In conclusion, evodiamine at a dose of 10 mg/kg/day appears to confer a protective effect on atherosclerosis and has no evident toxic side effects in mice.

### 2.2. Evodiamine Reduces Atherosclerotic VSMC Proliferation and Migration in Mice

To verify whether evodiamine can inhibit the proliferation and migration of VSMCs in mice fed on a high fat diet for 8 weeks, double immunofluorescence staining was performed in aortic root sections of mice. PCNA is a proliferative, and Myh11 + PCNA + fluorescence represents the proliferation propensity of VSMCs. The proliferation levels of VSMCs were significantly reduced in the evodiamine-treated mice compared to the controls (Figure 2A,B). To verify this finding, aortic root sections co-labeled with Acta2, EdU, and DAPI were used to analyze VSMCs’ proliferation or cell numbers. The results confirmed that evodiamine treatment was associated with significant inhibition in the proliferation level of VSMCs (Figure 2D,E). In addition, matrix metallopeptidase 2 (MMP2) was used as a marker of cell migration; Acta2 + MMP2 + regions represent migration of VSMCs. As shown in Figure 2A,C, the degree of VSMC migration in aortic root lesions was significantly lower in the evodiamine treatment group than in the control group.

Together, these results confirm that evodiamine treatment was associated with a reduction in the occurrence and impaired development of atherosclerotic plaques by regulating the proliferation and migration of VSMCs.

### 2.3. Evodiamine Inhibits ox-LDL-Induced VSMC Proliferation and Migration

The inhibitory effect of evodiamine on the proliferation and migration of VSMCs was further validated in an in vitro atherosclerosis model. First, the optimal drug concentration was deduced by incubating cells in the presence of increasing evodiamine concentrations. Evodiamine was dissolved in DMSO and diluted in DMEM to 0.25, 0.5, 1, 2, and 4 μM final concentrations, and the above concentrations were used to treat MOVAS cells for 72 h. A CCK8 cell activity test following the culture revealed that 1 μM or more of evodiamine affected cell viability (Figure 3A). Therefore, 0.5 μM evodiamine was used to perform the subsequent in vitro experiments.

We cultured the MOVAS cells with 80 mg/L ox-LDL for 72 h to induce VSMC proliferation and migration. Proliferation of MOVAS cells was significantly lower in the cells exposed to 0.5 μM evodiamine than in the control group (Figure 3B). In line with this, the EdU staining results confirmed that the number of EdU-positive cells was significantly reduced in evodiamine-exposed cells, indicating a decline in cell proliferation (Figure 3C,D). We measured the expression of the proliferation marker PCNA in whole protein lysate from each treatment group through Western blotting (Figure 3E). We confirmed that ox-LDL treatment induced PCNA protein expression in MOVAS cells. Furthermore, the PCNA protein abundance in ox-LDL-treated cells was significantly reduced in the evodiamine-treated cells (Figure 3F). Finally, RT-qPCR analysis revealed that the transcript levels of PCNA were lower in the evodiamine-treated cells, confirming that evodiamine treatment was associated with a reduction in *PCNA* mRNA (Figure 3J).

In addition to this proliferative analysis, we investigated the effect of evodiamine on MOVAS cell migration using a scratch test migration assay. Cells were cultured to confluence and then images were recorded at 0, 12, and 24 h after each scratch, and the area of blank space was quantified. The migration of MOVAS cells were significantly increased after ox-LDL exposure at both 12 h and 24 h (Figure 3N,O), while the migration levels of evodiamine-treated MOVAS cells were significantly impaired. The Western blot analysis revealed that the expression level of MMP2 protein, a migration marker, in ox-LDL-induced cells decreased significantly in the presence of evodiamine (Figure 3G). Furthermore, RT-qPCR confirmed that the transcript level of *MMP2* was lower in the evodiamine-exposed cells compared to the controls (Figure 3K).

Given that the proliferation and migration of VSMCs are often accompanied by cell phenotypic transformation during atherosclerosis, we also investigated the protein and mRNA expression levels of contraction phenotypic markers Acta2 and Myh11 in VSMCs. MOVAS cells had almost complete inhibition of expression of systolic contraction markers after ox-LDL exposure. However, evodiamine rescued expression of these markers (Figure 3E,H,I,L,M).

In conclusion, in vitro experiments further confirmed that evodiamine can delay the occurrence and development of atherosclerosis by inhibiting the conversion of VSMCs from a systolic phenotype to a synthetic phenotype and reduced the proliferation and migration levels of VSMCs.

### 2.4. Evodiamine Reduces Inflammation and Oxidative Stress in Mouse VSMCs

The atherosclerotic process is multi-faceted; both inflammation and oxidative stress exposure are major contributors to the development of atherosclerotic plaques. We aimed to determine whether evodiamine could regulate inflammation and oxidative stress in vivo. To address this, we performed immunofluorescent staining of mouse tissues to investigate the pattern of expression markers of inflammation and oxidative stress in mice in the presence or absence of evodiamine.

Double immunofluorescence staining was performed in mouse aortic root sections. Smooth muscle cell markers Acta2 or Myh11 were co-stained with inflammation-related markers TNF-α and IL-1β or oxidative stress markers Nrf2 and 4-HNE. These analyses revealed that the expression of the inflammatory markers was significantly reduced in VSMCs located within the plaques in mice treated with evodiamine and that Acta2 or Myh11 co-stained with the oxidative stress markers (Figure 4A–C). Furthermore, we observed that the expression levels of TNF-α and IL-1β in atherosclerotic VSMCs were significantly reduced in the evodiamine-treated mice. The expression levels of Nrf2 and 4-HNE in atherosclerotic VSMCs in the evodiamine-treated mice were remarkably lower compared with the control group (Figure 4D–F).

In summary, the anti-atherosclerotic properties of evodiamine appear to be mediated via regulation of inflammatory response and oxidative stress in VSMCs.

### 2.5. Evodiamine Attenuates the Inflammatory Response and Oxidative Stress in Atherosclerosis-Induced VSMCs by Inhibiting PI3K/Akt

To explore how evodiamine regulates atherosclerotic VSMCs’ inflammation and oxidative stress, we also conducted in vitro experiments using the MOVAS cell line. Firstly, the inhibitory effect of evodiamine on inflammation and oxidative stress of atherosclerotic VSMCs was verified. Both the Western blotting and RT-qPCR results demonstrated that the expression levels of inflammatory cytokines and oxidative-stress-related markers in smooth muscle cells were significantly lower in the evodiamine-treated cells compared with the controls (Figure 5A–H). Furthermore, ROS levels in atherosclerotic cell models were clearly decreased in the evodiamine-treated cells (Figure 5I,O).

In other diseases, such as pancreatic cancer, evodiamine has been shown to regulate PI3K/Akt activity. Moreover, studies have confirmed that the PI3K/Akt axis is involved in the regulation of inflammatory response and oxidative stress. Therefore, we determined the protein expression levels of p-PI3K, PI3K, p-Akt, and Akt in response to evodiamine exposure. The Western blotting results revealed that the expression levels of p-PI3K and p-Akt were decreased after evodiamine treatment; however, there was no statistically significant difference between the expression levels of total PI3K and total Akt (Figure 5J,K–N).

In summary, evodiamine may restrain the progression of atherosclerosis by inhibiting the activation of the PI3K/Akt axis in atherosclerotic VSMCs, reducing the level of inflammatory response and oxidative stress in these cells.

## 3. Materials and Methods

### 3.1. Animals

*LDLR* knockout (*LDLR−/−*) mice were provided by Changzhou Cavens (Cavens, Changzhou, China). To eliminate the effects of sex hormones from the experiment, only male mice were used in the experiment. To induce hypercholesterolemia, 8-week-old *LDLR−/−* mice were fed a high-fat diet containing 1–1.25% cholesterol (TP28640, Trophic, Nantong, China) for 8 weeks. One group was treated with 10 mg/kg/d by daily gavage evodiamine (Selleck, Houston, TX, USA) for 8 weeks. Evodiamine was dissolved in dimethyl sulfoxide (DMSO; Sigma, Abitibi belt, Canada), stored at −20 °C and diluted before use (2% DMSO + 30% PEG400 + saline). The control group was treated with saline. All animal experimental procedures were reviewed and sanctioned by the Animal Ethics Committee of Yangzhou University.

### 3.2. Cell Culture and Reagents

Mouse aortic vascular smooth muscle cells (MOVAS) were purchased from SAINT-BIO (SAINT-BIO, Shanghai, China) and cultured in high-glucose DMEM (HyClone, Logan, UT, USA) supplemented with 10% fetal bovine serum (Gibco, Grand Island, NY, USA) and 1% penicillin/streptomycin (Beyotime, Shanghai, China). Cells in good condition at the logarithmic growth stage were selected, digested with 0.25% trypsin (Beyotime, Shanghai, China), centrifuged, the supernatant removed, re-suspended, and inoculated in cell culture plates for the corresponding experiment. After 24 h of adherence, the complete medium was removed, and cells were cultured for another 24 h with serum starvation. Then, the cells were divided into control, ox-LDL (80 mg/L, Yiyuan Biotech, Guangzhou, China) treatment, and evodiamine (Selleck, Houston, TX, USA) treatment groups. Evodiamine was dissolved in DMSO at 10 μM. The final concentration of DMSO in the culture medium was maintained at 0.1%, which showed no cytotoxicity to the MOVAS.

### 3.3. Measurement of Plaque Morphology

After 8 weeks, the mice were anesthetized with isoflurane (3% + oxygen) during final tissue collection and then euthanized with isoflurane overdose (greater than 5% isoflurane concertation until 1 min after breathing stops). The whole aortas were opened longitudinally from the ascending aorta to the iliac bifurcation, pinned en face, and stained for lipids with Sudan 4 (Solarbio, Beijing, China). The stained area was identified as the atherosclerotic lesion and evaluated as a percentage of the total aortic area using ImageJ software. Hearts were fixed and embedded in paraffin to prepare serial cross-sections (5 mum thick) throughout the entire aortic root area for histological analysis. Paraffin sections of the aortic roots were stained with hematoxylin and eosin (H&E; Beyotime, Shanghai, China) and Masson (Solarbio, Beijing, China); in addition, collagen content, necrotic core, and plaque size were assessed using ImageJ software.

### 3.4. Cell Counting Kit-8 and 5-Ethynyl-2′-Deoxyuridine Assay

The cell growth was estimated by Cell Counting Kit-8 (CCK8) assay (Beyotime, Shanghai, China) according to the manufacturer’s instructions. MOVAS cells were plated in 96-well plates with 1 × 10^3^ cells per well. After 72 h of treatment, a 10 μL aliquot of CCK-8 reagent was added and incubated for a further 2 h at 37 ℃. Then, optical density (wavelength: 450 nm) was determined by an AMR-100 automatic enzyme label analyzer (Allsheng, Hangzhou, China).

Cell proliferation was measured by 5-ethynyl-2′-deoxyuridine (EdU) incorporation assay (Beyotime, Shanghai, China). After drug treatment, an EdU working solution (10 mM, with green fluorescence) was added to the medium and then incubated for 2 h. After incubation, the cells were fixed with 4% PFA, washed with PBS, permeabilized with 0.3% Triton X-100, added to a freshly prepared Click reaction solution, and the cells were then stained with Hoechst 33342.

### 3.5. RNA Isolation and qRT-PCR

For qRT-PCR, total RNA was extracted using an RNA-easy isolation reagent (Vazyme, Nanjing, China) and then reverse transcribed to cDNA with a HiScript III RT SuperMix for qPCR kit (Vazyme, Nanjing, China) at 37 ℃ for 15 min and 85 ℃ for 5 s. Quantitative PCR was performed using a ChamQ Universal SYBR qPCR master mix (Vazyme, Nanjing, China) and an Applied Biosystems QuantStudio 3 Detection System (Thermo Fisher Scientific, Waltham, MA, USA). The following primers were used:

*PCNA*: forward, 5′-TTGCACGTATATGCCGAGACC-3′; reverse, 5′-GGTGAACAGGCTCATTCATCTCT-3′.

*MMP2*: forward, 5′-ACCTGAACACTTTCTATGGCTG-3′; reverse, 5′-CTTCCGCATGGTCTCGATG-3′.

*Acta2*: forward, 5′-CCCAGACATCAGGGAGTAATGG-3′; reverse, 5′-TCTATCGGATACTTCAGCGTCA-3′.

*Myh11*: forward, 5′-AAGCTGCGGCTAGAGGTCA-3′; reverse, 5′-CCCTCCCTTTGATGGCTGAG-3′.

*Nrf2*: forward, 5′-CTTTAGTCAGCGACAGAAGGAC-3′; reverse, 5′-AGGCATCTTGTTTGGGAATGTG-3′.

*TNF-α*: forward, 5′-AACTCCAGGCGGTGCCTATG-3′; reverse, 5′-TCCAGCTGCTCCTCCACTTG-3′.

*IL-1β*: forward, 5′-AGCTTCAGGCAGGCAGTATC-3′; reverse, 5′-TCATCTCGGAGCCTGTAGTG-3′.

*GAPDH*, forward, 5′-TGGCCTTCCGTGTTCCTAC-3′; reverse, 5′-GAGTTGCTGTTGAAGTCGCA-3′ (Sangon Biotech, Shanghai, China).

### 3.6. Western Blotting and Immunofluorescence Antibodies

The following antibodies were used for Western blotting and immunofluorescence analyses: anti-GAPDH (CWBIO, Beijing, China, CW0100 M; WB, 1:1000), anti-Acta2 (HUABIO, Hangzhou, China, ER1003; IF, 1:100; WB, 1:1000), anti-Myh11 (Proteintech, Wuhan, China, 60222-1-lg; IF, 1:100; WB, 1:5000), anti-PCNA (HUABIO, Hangzhou, China, ET1605-38; IF, 1:100; WB, 1:1000), anti-MMP2 (Thermo Fisher Scientific, Shanghai, China, 436000; IF, 1:100; WB, 1:1000), anti-4-HNE (R & D Systems, Shanghai, China, MAB3249; IF, 1:50; WB, 1:1000), anti-Nrf2 (Abcam, Cambridge, UK, ab31163; IF, 1:100; WB, 1:1000), anti-TNF-α (Santa Cruz Biotechnologies Inc., CA, USA, sc-52746; IF, 1:200; WB, 1:1000), anti-IL-1β (Santa Cruz Biotechnologies Inc., CA, USA, sc-515598; IF, 1:200; WB, 1:1000), anti-PI3K (CST, Shanghai, China, #4249; WB, 1:1000), anti-p-PI3K (CST, Shanghai, China, #17366; WB, 1:1000), anti-Akt (CST, Shanghai, China, #4685; WB, 1:1000), anti-p-Akt (CST, Shanghai, China, #4060; WB, 1:1000).

### 3.7. EdU Detection in Tissue

The *LDLR−/−* mice were fed a high-fat diet (HFD) for 8 weeks, with or without evodiamine treatment, and EdU (50 mg/kg, Beyotime, Shanghai, China) was intraperitoneally injected for 4 h. After being anesthetized with 5% isoflurane for 5 min, and 1 min after breathing stopped, the mice were euthanized by decapitation. The aortic root tissue was fixed in 4% PFA and embedded in paraffin. Paraffin sections (5 μm) were dewaxed and dehydrated. Antigen retrieval was performed using citrate buffer; samples were then permeabilized and blocked in 3% bovine serum albumin (BSA) at room temperature for 1 h. The sections were incubated with primary antibodies in the blocking buffer overnight at 4 °C, and immunofluorescence staining was performed with the corresponding secondary antibody (1:500) (Thermo Fisher Scientific, Waltham, MA, USA). The EdU reaction solution was prepared accordingly using the manufacturer’s instructions (BeyoClick™ EdU-488 Cell Proliferation kit, Beyotime, Shanghai, China). The tissues were then incubated with the EdU reaction solution for 30 min at room temperature and protected from light.

The slides were stained with Hoechst 33,342 and washed with 3% BSA-PBS. Images were acquired with an AxioScopeA1 light microscope (Carl Zeiss, Jena, Germany).

### 3.8. Detection of Reactive Oxygen Species

Reactive oxygen species (ROS) were measured using a AAT Bioquest Fluorimetric Intracellular Total ROS Activity Assay Kit (#22,901, Sunnyvale, CA, USA). ROS red is cell-permeable and produces red fluorescence when reacting with ROS. The ROS red/DMSO mixture (20 μL) was added to 10 mL of assay buffer (component B of the kit) and mixed well to create a working solution. The working solution (100 μL) was added into each well of cells. The plate containing cells was then incubated for 90 min at 37 °C. Images were acquired with an AxioScopeA1 light microscope (Carl Zeiss, Jena, Germany).

### 3.9. Wound Healing Assay

A wound healing assay was used to analyze the MOVAS migration rate. In brief, the MOVAS were seeded into each well of 6-well plates and grown to confluence. After 24 h of serum deprivation, three parallel wounds with similar widths (<3 mm) were created in each well by using a sterile 200 µL pipette tip (this event was designated as the beginning of the experiment i.e., 0 h), and ox-LDL (80 mg/L) was added into the wells for stimulation. The cells were then treated with evodiamine or PBS. Through direct microscopic visualization, a reference point at the bottom was created in each field of the wound at 0 h, and wound closure rates were analyzed by photographing and measuring the remaining cell-free area in the identical field immediately and after 12 h and 24 h of stimulation.

### 3.10. Statistical Analysis

The results are shown as mean ± standard error of mean (SEM). All data represent independent data points and not technical replicates. Data with multiple predicting factors were analyzed by one- or two-way ANOVA with Tukey correction for multiple comparisons. Student’s t-tests (unpaired) were used to compare differences between two groups. GraphPad Prism 8.0 (GraphPad Software, La Jolla, CA, USA) was used for statistical analysis. Statistical significance was defined as *p* < 0.05.

## 4. Discussion

Evodiae fructus, the dried fruit of *Evodia rutaecarpa (Juss.) Benth*, is one of the traditional versatile herbs used in China to treat headaches, abdominal pain, vomiting, diarrhea, and other diseases [23,24,25]. Like other Chinese herbal medicine, Evodiae fructus has been reported to be relatively safe in the book “*Shen Nong’s Herbal Classic*”, the oldest book documenting the use of traditional Chinese medicinal herbs [26]. Evodiamine is an indole alkaloid as determined by phytochemical studies [27,28]. A growing body of evidence suggests that evodiamine is an important compound with a wide range of biological activities, such as anti-inflammatory [29,30], anti-tumor [31], and anti-microbial activities [32]. Evodiamine may interact with many different targets to achieve its therapeutic effect [33,34]. It is for this reason that evodiamine is gaining scientific interest from a medicinal perspective.

Additionally, evodiamine has been shown to have a beneficial effect on cardiovascular diseases in recent studies. It has been reported that evodiamine suppresses hypoxia-induced inflammatory responses in macrophages by inhibiting hypoxia-induced factor-alpha (HIF-α) [35]. Based on mechanistic studies, evodiamine directly binds ABC transporter A1 (ABCA1), increasing ABCA1 protein stability and promoting cholesterol efflux from macrophages [36]. Furthermore, evodiamine has been reported to prevent endothelial-to-mesenchymal transition (EndoMT) and isoproterenol-induced cardiac fibrosis [37]. It appears that evodiamine may reduce the risks associated with cardiovascular disease.

There are several causes of cardiovascular disease, which is a leading cause of mortality worldwide [38,39]. Atherosclerosis is the primary risk factor for cardiovascular disease [40,41,42]. Despite this, few studies have been conducted on the anti-atherosclerotic effects of evodiamine. We found that evodiamine treatment was associated with reduced atherosclerotic plaque formation in *LDLR−/−* mice fed a high-fat diet. It has been demonstrated that evodiamine reduces the size of atherosclerotic plaque by regulating proliferation, migration, inflammatory response, and oxidative stress in VSMCs. In other diseases, such as pancreatic cancer, the PI3K/Akt axis is the upstream signaling pathway through which evodiamine modulates cellular activity [43]. Moreover, studies have also confirmed that the PI3K/Akt axis is involved in the regulation of inflammatory response and oxidative stress [44,45,46]. Following evodiamine treatment, p-PI3K and p-Akt protein levels decreased in vitro. Therefore, it can be concluded that evodiamine may inhibit the activation of the PI3K/Akt axis in atherosclerotic VSMCs and reduce the level of inflammatory response and oxidative stress, thus delaying the process of atherosclerosis.

In spite of the fact that evodiamine inhibits the occurrence and development of atherosclerotic plaques, it is unable to completely prevent plaque formation. Consequently, more efforts may be needed to improve the bioavailability of evodiamine. Evodiamine is not particularly soluble, and some studies have shown that the solid dispersion of evodiamine in hard capsules has a higher absorption rate than concentrated evodiamine samples in physical mixtures of hard capsules [47]. In another study, a novel evodiamine-phospholipid complex with higher bioavailability than evodiamine was described [48]. Given the potential of evodiamine as a therapy for atherosclerosis, the mechanisms behind its anti-atherosclerotic effect, improving its bioavailability, and assessing its clinical efficacy and side effects are all priority research questions.

## 5. Conclusions

In conclusion, evodiamine can effectively reduce atherosclerotic plaque formation in mice, and VSMCs play an important role in this process. Studies on the underlying mechanisms demonstrate that evodiamine can reduce the proliferation and migration of VSMCs in atherosclerotic plaques via regulation of the inflammatory response and oxidative stress in VSMCs through the PI3K/Akt axis. Together, these interventions appear to postpone the progression of atherosclerosis.

## Figures and Tables

**Figure 1 ijms-24-06653-f001:**
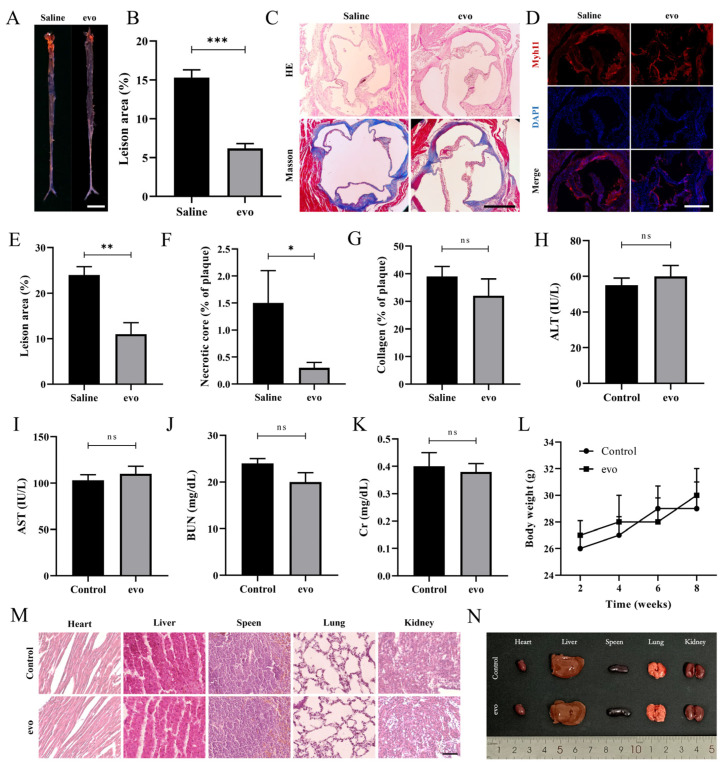
Analysis of aortic plaques and evodiamine toxicology in mice: (**A**,**B**) images taken of aortic Sudan 4 staining and quantification. Scale bars in (**A**), 1 cm. (**C**, **E**–**G**) The paraffin sections of aortic roots were stained with H&E and Masson and the stained regions were quantified. (**D**) The cross-section of the aortic root was analyzed by immunofluorescence staining. (**H**–**L**) Liver and kidney function tests and weight measurements. Scale bars in (**C**,**D**), 500 μm. (**M**,**N**) H&E staining and appearance of organs. Scale bars in (**M**), 50 μm. The number of mice in each group was six. The bar graphs depict means ± SEM; statistical significance * *p* < 0.05, ** *p* < 0.01, *** *p* < 0.001 and ns *p* > 0.05 by the one-way ANOVA test.

**Figure 2 ijms-24-06653-f002:**
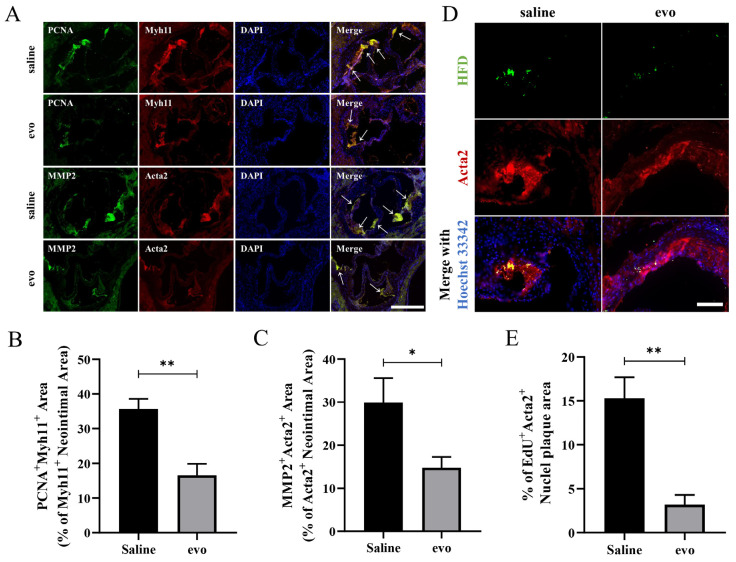
Analysis of proliferation and migration of atherosclerotic VSMCs in mice: (**A**–**C**) the transverse section of the aortic root was analyzed and quantified by immunofluorescence staining. PCNA and MMP2 expression in atherosclerotic plaques (green). Immunofluorescence shows that Myh11 (red) co-express PCNA and Acta2 (red) co-express MMP2 within the atherosclerotic lesion. The white arrow points to the co-stained area. Scale bar 500 μm. (**D**,**E**) Representative EdU staining and quantification of proliferating cells in plaques. Yellow fluorescence indicates EdU + Acta2 + cells. Scale bar 100 μm. The bar graphs depict means ± SEM; statistical significance * *p* < 0.05 and ** *p* < 0.01 by the one-way ANOVA test.

**Figure 3 ijms-24-06653-f003:**
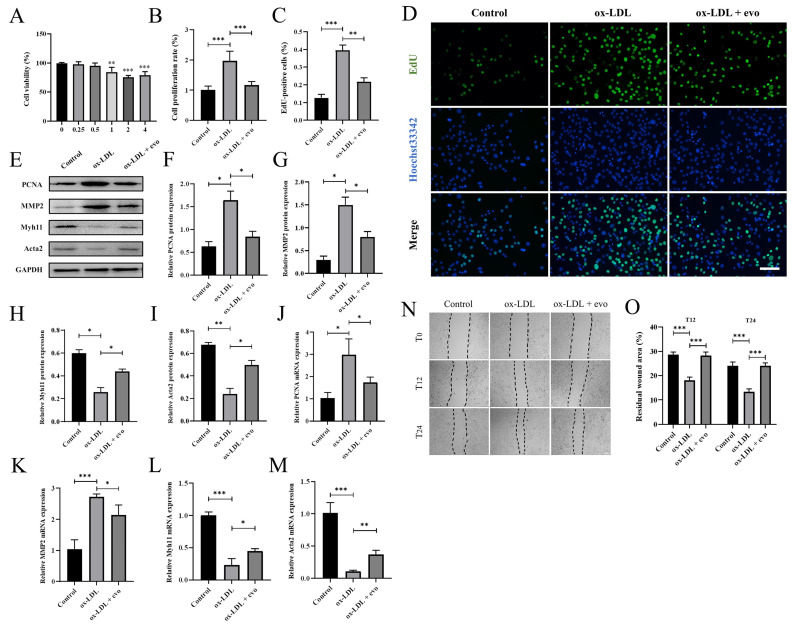
Analysis of proliferation and migration of atherosclerotic VSMCs in vitro: (**A**) the effect of different concentrations of evodiamine on cell viability was analyzed by CCK-8 assay. (**B**) Calculation of cell proliferation rate. (**C**,**D**) VSMC proliferation was quantified by EdU assay. Scale bar 100 μm. (**E**–**I**) Representative images (**E**) and data of the protein levels of PCNA (**F**), MMP2 (**G**), Myh11 (**H**) and Acta2 (**I**). (**J**–**M**) The expression of *PCNA*, *MMP2*, *Myh11*, and *Acta2* was quantified by qPCR and standardized to *GAPDH*. (**N**,**O**) Scratch test and blank area statistics. Scale bar 100 μm. The bar graphs depict means ± SEM; statistical significance * *p* < 0.05, ** *p* < 0.01, and *** *p* < 0.001 by the one-way ANOVA test.

**Figure 4 ijms-24-06653-f004:**
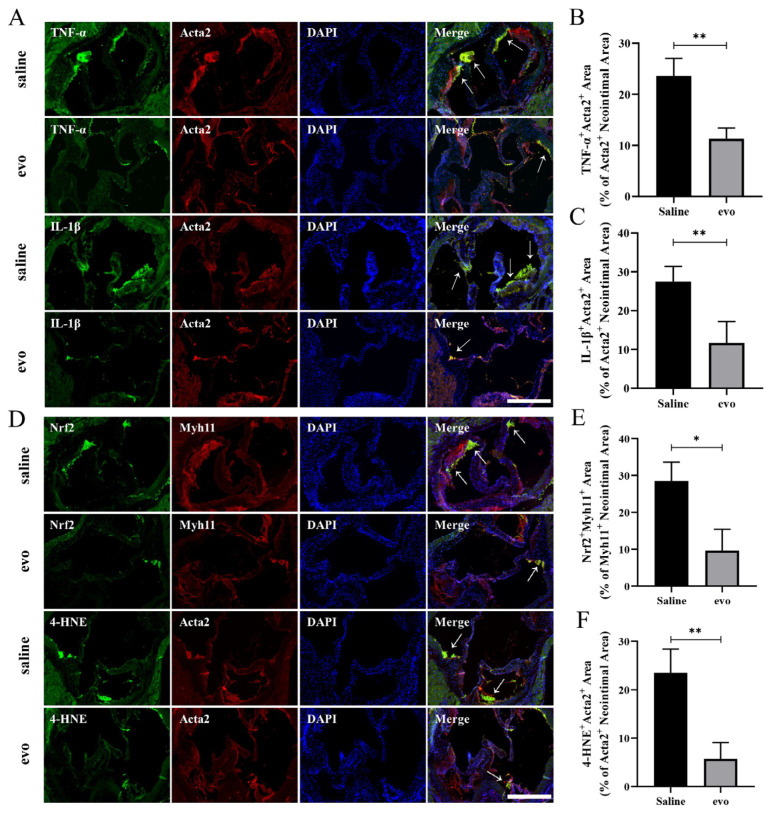
Analysis of inflammatory response and oxidative stress levels of atherosclerotic VSMCs in mice: (**A**–**C**) the transverse section of the aortic root was analyzed and quantified by immunofluorescence staining. TNF-α and IL-1β expression in atherosclerotic plaques (green). Immunofluorescence shows that Acta2 (red) co-expresses TNF-α and IL-1β within the atherosclerotic lesion. The white arrow points to the co-stained area. Scale bar 500 μm. (**D**–**F**) Nrf2 and 4-HNE expression in atherosclerotic plaques (green). Immunofluorescence shows that Myh11 (red) co-expresses Nrf2 and Acta2 (red) co-expresses 4-HNE within the atherosclerotic lesion. The white arrow points to the co-stained area. Scale bar 500 μm. The bar graphs depict means ± SEM; statistical significance * *p* < 0.05 and ** *p* < 0.01 by the one-way ANOVA test.

**Figure 5 ijms-24-06653-f005:**
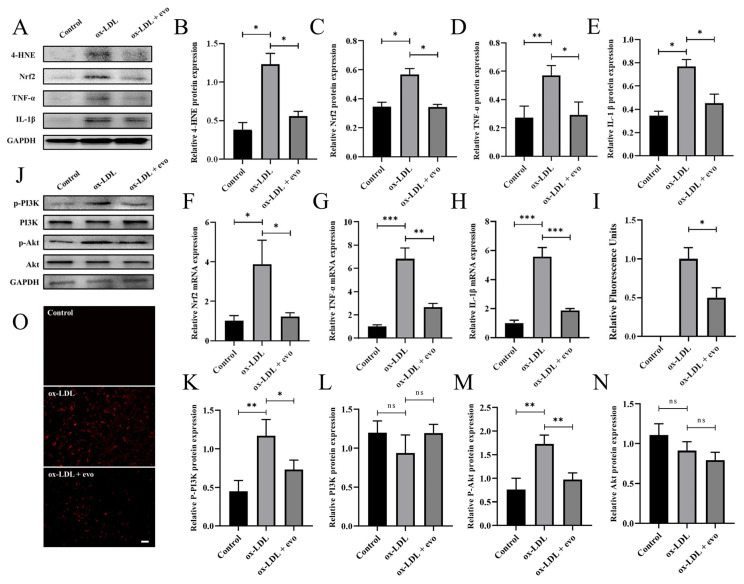
Analysis of inflammatory response and oxidative stress levels of atherosclerotic VSMCs in mice: (**A**–**E**) representative images (**A**) and data of the protein levels of 4-HNE (**B**), Nrf2 (**C**), TNF-α (**D**), and IL-1β (**E**). (**F**–**H**) The expression of *Nrf2*, *TNF-α*, and *IL-1β* was quantified by qPCR and standardized to GAPDH. (**I**,**O**) ROS fluorescence detection (red) and quantification. Scale bar 100 μm. (**J**–**N**) Representative images (**J**) and data of the protein levels of P-PI3K (**K**), PI3K (**L**), P-Akt (**M**), and Akt (**N**). The bar graphs depict means ± SEM; statistical significance * *p* < 0.05, ** *p* < 0.01, and *** *p* < 0.001 and ns by the one-way ANOVA test.

## Data Availability

The study did not report any data.

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
