# Peer review of "Dietary Evodiamine Inhibits Atherosclerosis-Associated Changes in Vascular Smooth Muscle Cells"

_ijms, 2023, doi:10.3390/ijms24076653_

Round 1
Reviewer 1 Report
The manuscript by Zha et al. investigates Dietary Evodiamine in a mouse model of atherosclerosis. Using the widely accepted LDLR-/- mouse model, this manuscript reports significant reductions in plaque burden in the aorta and aortic root and changes in vascular smooth muscle cell phenotype. This manuscript further characterizes this phenotype in vitro using a MOVAS cell line. The work provides preliminary evidence for Evodiamine partially restoring a contractile phenotype in vascular smooth muscle cells and reducing atherosclerotic burden.
As this manuscript is well conceptualized regarding experimental design and overall flow, I believe it will interest a broad audience. However, significant attention to the composition of this manuscript is required before it is of suitable quality for publication or review.
Briefly, the figure quality is not up to acceptable standards. All images in this manuscript are not of high enough resolution to be accurately viewed or interpreted. In addition, multiple editing errors appear on the figures. Figures 2-5 have graphs or images covered by information boxes detailing image size. For example, figure 3O has a green line through the graph, and figures 2A, 3G, and 5A have an information box detailing the image size covering them. Furthermore, the figure legend for image 1 does not reference Figure 1M or N, and figure legend 2 is duplicated in the text.
Author Response
Dear Editors and Reviewers:
Thank you for giving us an opportunity to revise our manuscript. We appreciate the editor and reviewers very much for their constructive comments and suggestions on our manuscript entitled "Dietary evodiamine inhibits atherosclerosis-associated changes in vascular smooth muscle cells" ( ijms-2281737).
We have studied reviewers’ comments carefully. According to the reviewers’ detailed suggestions, we have made a careful revision on the original manuscript. All revisions to the manuscript are marked up using the “Track Changes” function which we would like to submit for your kind consideration.
Should you have any other questions, please feel free to contact me.
Yours sincerely,
Yiwen Zha
We would like to express our sincere thanks to the editor and reviewers for the constructive and positive comments.
Replies to Reviewer #1
Specific Comments:
Question 1: The figure quality is not up to acceptable standards. All images in this manuscript are not of high enough resolution to be accurately viewed or interpreted. Multiple editing errors appear on the figures. Figures 2-5 have graphs or images covered by information boxes detailing image size. For example, figure 3O has a green line through the graph, and figures 2A, 3G, and 5A have an information box detailing the image size covering them.
Response: Thank you very much for your comments. We have improved the quality of the figures and made corrections in the corresponding parts. (Figure 2-5)
Question 2: The figure legend for image 1 does not reference Figure 1M or N, and figure legend 2 is duplicated in the text.
Response: Figure 1M or N has been added to the Figure legend for image 1 (Line 224). Duplicate figure legend 2 has been removed (Line 228-235).
Special thanks to you for your good comments.
Reviewer 2 Report
In this paper, the authors evaluated the effect of evodiamine on the vascular smooth muscle cells properties (VSMCs) and consequently its potential anti-atherosclerotic effect. Both animal and cell models are used to explain the molecular mechanism.
It is not clear how evodiamine is used in traditional Chinese medicine.
Evodiamine is known as a very toxic molecule. Mice received 10 mg/kg/day of this compound. How this quantity has been determined?
Line 404 : authors wrote that "evodiamine is not particularly soluble". What is the solvant used for the preparation of the solution for the gavage? And for culture cell, how the authors make sure that the molecule is well dissolved in the culture cell medium?
Description of animal experiment is incomplete : nothing is mentioned about the treatment of the different analyzed tissues.
L78 : "without serum starvation" should be "with serum starvation"?
L95 : 10^3 instead of 103
L213 : figure 1 : the legend for M,N is missing
L249 : Figure 2 : "B" or "D"?
Conclusion should come after Discussion
Author Response
Dear Editors and Reviewers:
Thank you for giving us an opportunity to revise our manuscript. We appreciate the editor and reviewers very much for their constructive comments and suggestions on our manuscript entitled "Dietary evodiamine inhibits atherosclerosis-associated changes in vascular smooth muscle cells" ( ijms-2281737).
We have studied reviewers’ comments carefully. According to the reviewers’ detailed suggestions, we have made a careful revision on the original manuscript. All revisions to the manuscript are marked up using the “Track Changes” function which we would like to submit for your kind consideration.
Should you have any other questions, please feel free to contact me.
Yours sincerely,
Yiwen Zha
We would like to express our sincere thanks to the editor and reviewers for the constructive and positive comments.
Replies to Reviewer #2
Specific Comments:
Question 1: It is not clear how evodiamine is used in traditional Chinese medicine.
Response: For thousands of years, Evodia rutaecarpa, the plant rich in evodiamine and rutaecarpine, has been extensively applied as central agent in classic TCM formulas (Wu-zhu-yu decoction and Zuo-jin-wan) in the treatment of migraine (termed ‘Jueyin headache’, in ancient China.) and other diseases. Thank you very much for your advice, we have added relevant additions in the introduction section (Line 42-44).
Question 2: Evodiamine is known as a very toxic molecule. Mice received 10 mg/kg/day of this compound. How this quantity has been determined?
Response: Evodiamine is indeed toxic, but this only happens when high concentrations and high doses are used [DOI: 10.1124/dmd.117.080176 (80mg/kg); DOI: 10.1016/j.jep.2013.12.006 (667mg/kg/d and 2000mg/kg/d)]. Evodiamine has been shown to be mildly toxic at concentrations of 30µM in vitro [DOI: 10.1055/s-2007-993743]. Besides, evodiamine is well known for its anti-tumor effect. In most oncology studies, the conventional dose in mice is 20mg/kg/d (or more, up to 40mg/kg/d) [DOI: 10.1007/s13277-016-5251-3; DOI: 10.1016/j.intimp.2015.07.030]. For the study of non-neoplastic disease, the dose of 10mg/kg/d was selected [DOI: 10.1002/jcb.28976; DOI: 10.1111/apha.12005]. Therefore, we referred to previous research methods and administered the drug at a concentration of 10mg/kg/d. Finally, the results showed that the use of this concentration of evodiamine had no side effect on mice.
Question 3: Line 404 : authors wrote that "evodiamine is not particularly soluble". What is the solvant used for the preparation of the solution for the gavage? And for culture cell, how the authors make sure that the molecule is well dissolved in the culture cell medium?
Response: Evodiamine is insoluble in solvents such as water and ethanol but can be dissolved in DMSO. Therefore, evodiamine was dissolved in DMSO for preservation and then diluted for use. In vivo: 2% DMSO + 30% PEG400 + saline. In vitro: 0.1% DMSO + culture cell medium. I'm sorry that we omitted this important part when writing the manuscript. We have supplemented the details. (Line 69-71 , 83-85 and 264)
Question 4: Description of animal experiment is incomplete : nothing is mentioned about the treatment of the different analyzed tissues.
Response: We have added the treatment methods of longitudinal section of mouse aorta and transverse section of aorta root in Line 93 and 94.
Question 5: L78 : "without serum starvation" should be "with serum starvation"?
Response: We have corrected the error. (Line 81)
Question 6: L95 : 10^3 instead of 103
Response: We have replaced 103 with 10^3. (Line 102)
Question 7: L213 : figure 1 : the legend for M,N is missing
Response: Figure 1M or N has been added to the Figure legend for image 1. (Line 224)
Question 8: L249 : Figure 2 : "B" or "D"?
Response: We have replaced "D" with "B". (Line 256)
Question 9: Conclusion should come after Discussion
Response: We have put Conclusion after Discussion. (Line 375-425)
Special thanks to you for your good comments.
Round 2
Reviewer 2 Report
Dear author,
Thank you for your answers.
I have no more comments.